# GNAT: A General Narrative Alignment Tool

**Tanzir Pial, Steven Skiena**
Department of Computer Science,
Stony Brook University, NY, USA
{tpial,skiena}@cs.stonybrook.edu

## Abstract

Algorithmic sequence alignment identifies similar segments shared between pairs of documents, and is fundamental to many NLP tasks. But it is difficult to recognize similarities between distant versions of narratives such as translations and retellings, particularly for summaries and abridgements which are much shorter than the original novels.

We develop a general approach to narrative alignment coupling the Smith-Waterman algorithm from bioinformatics with modern text similarity metrics. We show that the background of alignment scores fits a Gumbel distribution, enabling us to define rigorous p-values on the significance of any alignment. We apply and evaluate our general narrative alignment tool (GNAT) on four distinct problem domains differing greatly in both the relative and absolute length of documents, namely summary-to-book alignment, translated book alignment, short story alignment, and plagiarism detection—demonstrating the power and performance of our methods.

## 1 Introduction

Algorithmic sequence alignment is a fundamental task in string processing, which identifies similar text segments shared between a pair of documents. Sequence alignment is a common operation in many NLP tasks, with representative applications including identifying spelling (Dargis et al., 2018) and OCR (Yalniz and Manmatha, 2011) errors in documents, quantifying post-publication edits in news article titles (Guo et al., 2022), and plagiarism detection (Potthast et al., 2013). Text alignment has been used to create datasets for text summarization tasks (Chaudhury et al., 2019), by loosely aligning summary paragraphs to the chapters of original stories and documents. Textual criticism is the study of the transmission of text (Abbott and Williams, 2014), typically for religious and historically significant texts such as the Bible. The collation task in textual criticism examines textual variations across different versions of a text through alignment (Yousef and Janicke, 2020). Text alignment is often deployed in literary research, e.g., analyzing how books are adapted for young adults (Sulzer et al., 2018), translated (Bassnett, 2013), and how the gender of the translator affects the translated work (Leonardi, 2007).

However, sequence alignment in NLP research is largely done on an ad hoc basis, serving small parts of bigger projects by relying on hand-rolled tools. This is in contrast to the field of bioinformatics, where the alignment of nucleotide and protein sequences plays a foundational role. Popular tools such as BLAST (Altschul et al., 1990) running on large sequence databases are used to identify even distant homologies (similarites) with rigorous statistical measures of significance. Such tools facilitate meaningful analyses over vast differences in scale, from short gene-to-gene comparisons to full genome-to-genome alignment or gene-to-database searches.

The overarching goal of our work is to extend the rigorous sequence analysis techniques from bioinformatics to the world of narrative texts. Sequence alignment can be computed using the widely-used edit distance algorithm for Leveinshtein disance (Levenshtein et al., 1966), which aligns texts by computing the minimum number of deletions, insertions, substitutions, and/or transpositions. However, edit distance fails when applied to distant narrative texts that are semantically but not textually similar, since neither character nor word-level changes capture the semantic meaning of the text. Consider comparing two independent translations of a particular novel, or two different retellings of a classic fable. We anticipate very little in terms of long common text matches, even though the documents are semantically identical, and it is not obvious how to quantify the significance of whatever matches we do happen to find.

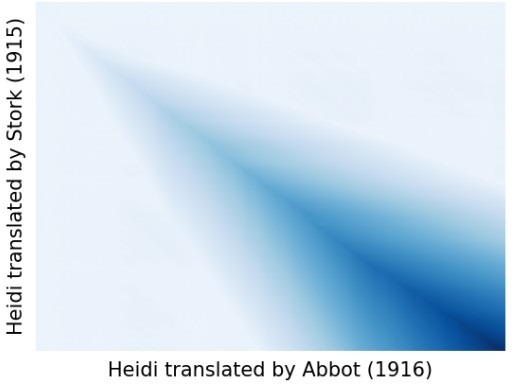

(a) Heidi Alignment by SW + SBERT

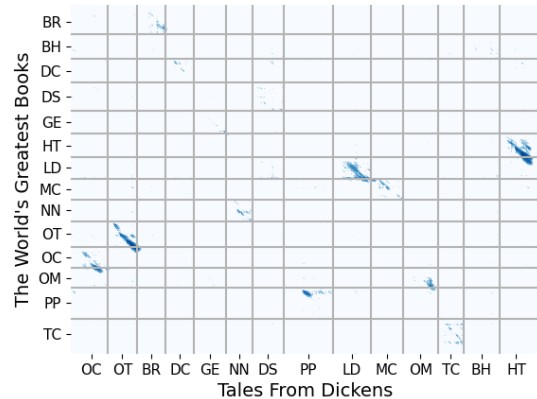

(b) Aligning versions of Dickens' tales by SW + SBERT

Figure 1: Heat maps representing Smith-Waterman alignments in two distinct narrative domains. On left, two different English translations of the novel *Heidi* are aligned: the bright main diagonal correctly indicates that they follow the same sequence of events. On right, two different story collections of abridged novels are aligned: the disjoint diagonal patterns correctly identify different versions of the same novels (represented using acronyms) shared across the collections.

Aligning large text documents is a difficult task, for reasons beyond measuring distance in semantic space versus simple text-based edits. Often the proper interpretation consists of multiple local alignments instead of one sequential global alignment. For instance, consider a document that plagiarizes disjoint parts of a second document in a different order. While local sequence alignments are successfully used in the bioinformatics domain to align genome and protein sequences (Smith et al., 1981), they have been underutilized in NLP. A second issue is scale mismatch: aligning a short book summary to an unabridged novel requires mapping single sentences to pages or even chapters of the larger text. A final concern is the statistical rigor of an alignment: every pair of completely unrelated texts has an optimal alignment, but how can we tell whether such an alignment exhibits meaningful measures of similarity?

Although text alignment is used in a NLP tasks across multiple domains, much of the existing work has been domain-specific or focused on global alignments. In this paper, we develop and evaluate a general purpose tool (GNAT) for the efficient and accurate alignment of pairs of distant texts. We propose a method for adapting the classical Smith-Waterman (SW) local alignment algorithm with affine gap penalties for NLP tasks by employing multiple textual similarity scoring functions and perform comparative analysis.

Our major contributions[1][2] are as follows:

- *Local Alignment Methods for Narrative Texts* – We develop and evaluate sequence alignment methods for distant but semantically similar texts, and propose a general method for computing statistical significance of text alignments in any domain. Specifically, we demonstrate that alignment scores of unrelated pairs of narrative texts are well-modelled by a Gumbel distribution (Altschul and Gish, 1996). Fitting the parameters of this distribution provides a rigorous way to quantify the significance of putative alignments.

- *Distance Metrics for Narrative Alignment* – We propose and evaluate five distinct distance metrics for the alignment of narrative documents over a range of relative and absolute sizes. We demonstrate that neural similarity measures like SBERT generally outperform other metrics, although the simpler and more efficient Jaccard similarity measure proves surprisingly competitive on task like identifying related pairs of book translations (0.94 AUC vs. 0.99 AUC for SBERT). However, we show that Jaccard similarity loses sensitivity over larger (chapter-scale) text blocks.

---

[1]All codes and datasets are available at https://github.com/tanzir5/alignment_tool2.0

[2]An associated web interface can be found at https://www.aligntext.com/

The Dodger and Charley Bates had taken Oliver out for a walk, and after sauntering along, they suddenly pulled up short on Clerkenwell Green, at the sight of an old gentleman reading at a bookstall. So intent was he over his book that he might have been sitting in an easy chair in his study. To Oliver's horror, the Dodger plunged his hand into the gentleman's pocket, drew out a handkerchief, and handed it to Bates. Then both boys ran away round the corner at full speed. Oliver, frightened at what he had seen, ran off, too; the old gentleman, at the same moment missing his handkerchief, and seeing Oliver scudding off, concluded he was the thief, and gave chase, still holding his book in his hand. The cry of "Stop thief!" was raised. Oliver was knocked down, captured, and taken to the police-station by a constable. The magistrate was still sitting, and Oliver would have been convicted there and then but for the arrival of the bookseller.

You can imagine Oliver's horror when he saw him thrust his hand into the old gentleman's pocket, draw out a silk handkerchief and run off at full speed. In an instant Oliver understood the mystery of the handkerchiefs, the watches, the purses and the curious game he had learned at Fagin's. He knew then that the Artful Dodger was a pickpocket. He was so frightened that for a minute he lost his wits and ran off as fast as he could go. Just then the old gentleman found his handkerchief was gone and, seeing Oliver running away, shouted "Stop thief!" which frightened the poor boy even more and made him run all the faster. Everybody joined the chase, and before he had gone far a burly fellow overtook Oliver and knocked him down. A policeman was at hand and he was dragged, more dead than alive, to the police court, followed by the angry old gentleman.

Table 1: Excerpts from two distinct abridgements of *Oliver Twist*. The highlighted areas indicate pairs of segments aligned by our text alignment tool.

- *Performance Across Four Distinct Application Domains* –To prove the general applicability of GNAT, we evaluate it in four application scenarios with text of varying absolute and relative sizes:

  - *Summary-to-book alignment:* Our alignment methods successfully match the summary with the correct book from the set of candidates with 90.6% accuracy.
  - *Translated book alignment:* We have constructed a dataset of 36 foreign language books represented in Project Gutenberg by two independent, full-length translations into English. Our alignment methods achieve an AUC score of 0.99 in distinguishing duplicate book pairings from background pairs.
  - *Plagiarism detection:* Experiments on the PAN-13 dataset (Potthast et al., 2013) demonstrate the effectiveness of our alignment methods on a vastly different domain outside our primary area of interest. Our $F_1$ score of 0.85 on the summary obfuscation task substantially outperforms that of the top three teams in the associated competition (0.35, 0.46, and 0.61, respectively).
  - *Short story alignment:* We conduct experiments on a manually annotated dataset of 150 pairs of related short stories (Aesop's fables), aligning sentences of independently written versions of the same underlying tale. Sentence-level Smith-Waterman alignments using

SBERT ($F_1 = 0.67$) substanially outperform baselines of sequential alignments and a generative one-shot learner model (0.40 and 0.46, respectively).

This paper is organized as follows. Section 2 provides an overview of related works on sequence alignments in NLP and bioinformatics. Section 3 formally defines the problem of text alignment and presents our alignment method with the different metrics for scoring textual similarities. Section 4 details our method for computing the statistical significance of text alignments. We describe our experimental results in Section 5 before concluding with future directions for research in Section 6.

## 2 Related Work

The literature on sequence alignment algorithms and applications is vast. Here we limit our discussion to representative applications in NLP and bioinformatics, and prior work on measuring the statistical significance of alignments.

### 2.1 Algorithmic Sequence Alignment

Dynamic programming (DP) is a powerful technique widely used in sequence alignment tasks, closely associated with dynamic time warping (DTW) (Müller, 2007). Everingham et al. (2006) and Park et al. (2010) used DTW to align script dialogues with subtitles. Thai et al. (2022) used the Needleman-Wunstch DP algorithm with an embedding-based similarity measure to create pairwise global alignments of English translations of the same foreign language book. Apart from DP,

Naim et al. (2013) uses weighted $A^*$ search for aligning multiple real-time caption sequences.

## 2.2 Representative NLP Applications

Statistical Machine Translation (MT) (Lopez, 2008) and neural MT models (Bahdanau et al., 2014) learn to compute word alignments using large corpora of parallel texts. However, these models are limited to word-level alignment and face computational inefficiencies for longer texts (Udupa and Maji, 2006). Multiple works (Mota et al., 2016, 2019; Sun et al., 2007; Jeong and Titov, 2010) have proposed joint segmentation and alignment can create better text segmentations via aligning segments sharing the same topic and having lexical cohesion. Paun (2021) uses Doc2Vec embeddings for alignment and creates monolingual parallel corpus.

By using text alignment tools to automate the alignment process, researchers can streamline their analyses and uncover new insights in a more efficient manner. For example, Janicki et al. (2022) performs large scale text alignment to find similar verses from ∼90,000 old Finnic poems by using clustering algorithms based on cosine similarity of character bigram vectors. Janicki (2022) uses embeddings of texts and proposes optimizations for simpler modified versions of the DP-based Needleman-Wunstch algorithm.

Pial et al. (2023) extends the methods proposed for GNAT for book-to-film script alignment and do analysis on the scriptwriter's book-to-film adaptation process from multiple perspectives such as faithfulness to original source, gender representation, importance of dialogues.

Foltỳnek et al. (2019) categorizes the extrinsic plagiarism detection approach as aligning text between a pair of suspicious and source document. Potthast et al. (2013) notes that many plagiarism detection algorithms employ the seed-and-extend paradigm where a seed position is first found using heuristics and then extended in both directions. Plagiarism detection tools can align different types of text, including natural language document, source codes (Bowyer and Hall, 1999), and mathematical expressions (Meuschke et al., 2017).

## 2.3 Sequence Alignment in Bioinformatics

DNA and protein sequence alignment have been extensively used in bioinformatics to discover evolutionary differences between different species.

Smith et al. (1981) modified the global Needleman-Wunstch algorithm (Needleman and Wunsch, 1970) to compute optimal local alignments and created the Smith-Waterman (SW) algorithm. In this paper, we use the SW algorithm as our choice of alignment method, but adapt it to the NLP domain.

These algorithms are quadratic in both time and space complexity. Hirschberg (1975) and Myers and Miller (1988) brought down the space complexity of these algorithms to linear. Nevertheless, pairwise alignment between a query sequence and a vast database remains a computationally intensive task. Altschul et al. (1990) proposed a faster but less accurate algorithm called BLAST for protein searches in database. We argue that with appropriate similarity metrics and modifications, these algorithms can be applied to narrative alignment.

## 2.4 Statistical Significance of Alignments

For ungapped nucleotide alignments where a single alignment must be contiguous without gaps, Karlin and Altschul (1990) proposed an important method for computing the statistical significance of an alignment score. The distribution of scores of ungapped alignments between unrelated protein or genome sequences follows an extreme-value type I distribution known as Gumbel distribution (Ortet and Bastien, 2010). For gapped alignments, no such analytical method is available but many empirical studies have demonstrated that the gapped alignment scores also follow an extreme value distribution (Altschul and Gish, 1996), (Pearson, 1998), (Ortet and Bastien, 2010). Altschul and Gish (1996) proposes a method to estimate the statistical significance of gapped alignment scores empirically. We adapt and empirically evaluate these methods for computing the statistical significance of narrative alignments.

## 3 Methods for Alignment

Text alignment involves matching two text sequences by aligning their smaller components, such as characters, words, sentences, paragraphs, or even larger segments like chapters. The best component size depends on the task and the user's intent. For example, aligning a summary with a book requires comparing sentences from the summary with paragraphs or chapters, while aligning two book translations requires comparing paragraphs, pages, or larger units.

Formally, we define text alignment as follows:

given text sequences $X = (x_0, x_1, ..., x_{m-1})$ and $Y = (y_0, y_1, ..., y_{n-1})$, we seek to identify a set of alignments where an alignment $a = \langle x_i, x_j, y_p, y_q \rangle$ indicates that the text segment $x_i$ to $x_j$ corresponds to the text segment $y_p$ to $y_q$.

## 3.1 The Smith-Waterman (SW) Algorithm

Here we use the local alignment method proposed by Smith et al. (1981), defined by the recurrence relation $H(i, j)$ in Equation 1 that attempts to find the maximal local alignment ending at index $i$ and index $j$ of the two sequences.

$$H(i,j) = \max \begin{cases} H(i-1, j-1) + S(X_i, Y_j), \\ H(i-1, j) + g, \\ H(i, j-1) + g, \\ 0 \end{cases} \quad (1)$$

$S(a, b)$ is a function for scoring the similarity between components $a$ and $b$ and $g$ is a linear gap penalty. However, we employ the more general affine gap penalty (Altschul, 1998) with different penalties for starting and extending a gap as the deletion of a narrative segment is often continuous, therefore extending an already started gap should be penalized less than introducing a new gap. The time complexity of the SW algorithm is $O(mn)$, where $m$ and $n$ are the lengths of the two sequences. We discuss how we obtain multiple local alignments based on the DP matrix created by $SW$ in Apppendix A.

## 3.2 Similarity Scoring Functions

The key factor that distinguishes aligning text sequences from other types of sequences is how we define the similarity scoring function $S(a, b)$ in Equation 1. An ideal textual similarity function must capture semantic similarity. This section describes several similarity scoring functions as Section 5.2 evaluates their comparative performance.

**SBERT Embeddings:** Reimers and Gurevych (2019) proposed SBERT to create semantically meaningful text embeddings that can be compared using cosine similarity. SBERT creates embeddings of texts of length up to 512 tokens, which is roughly equivalent to 400 words (Gao et al., 2021). Usually this limit works for all sentences and majority of paragraphs. For texts longer than 400 words, we chunk the text into segments of 400 words and do a mean pooling to create the embedding following Sun et al. (2019). SBERT has previously

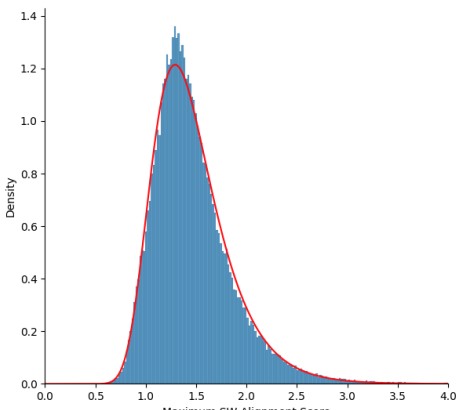

Figure 2: The distribution of maximum SW alignment scores from $2.5 \times 10^5$ pairs of unrelated books, where the red curve is the Gumbel distribution estimated from this data using maximum likelihood estimation (location $\mu = 1.29$, scale $\beta = 0.30$).

been used in computing semantic overlap between summary and documents (Gao et al., 2020).

**Jaccard:** The Jaccard index treats text as a bag-of-words, and computes similarity using the multiset of words present in both text segments:

$$J(a, b) = \frac{a \cap b}{a \cup b} \quad (2)$$

Jaccard has the caveat that it is not suitable for computing similarity between two texts of different lengths. Diaz and Ouyang (2022) has used Jaccard index for text segmentation similarity scoring.

**TF-IDF:** The Term Frequency Inverse Document Frequency (TF-IDF) measures text similarity using the frequency of words weighted by the inverse of their presence in the texts, giving more weight to rare words. Here the set of documents is represented by the concatenated set of text segments from both sequences. Chaudhury et al. (2019) uses TF-IDF to align summaries to stories.

**GloVe Mean Embedding**: Following Arora et al. (2017), we represent a text by the average of GloVe (Pennington et al., 2014) embeddings of all the words in the text, and compute cosine similarity.

**Hamming Distance:** Given two text segments containing $n \geq m$ words, we decompose the longer text into chunks of size $n/m$. The Hamming distance $h(.,.)$ is the fraction of chunks where chunk $i$ does not contain word $i$ from the shorter text. We then use $1 - h(.,.)$ as the similarity score.

## 3.3 Unifying Similarity Scoring Functions

The diverse ranges and interpretations of similarity scores from different scoring functions make

| | # of pairs | Total # of words | Avg. # of sentences per book/summary/fable | Avg. # of paragraphs per book/summary/fable | Human Annotation |
|---|---|---|---|---|---|
| RelBook | 36 | 6.56 Million | 4668 | 1953.7 | ✗ |
| Classics Stories | 14 | 146.9K | 271.1 | 114.5 | ✗ |
| (S)ummary(B)ook | 464 | 542K (S) | 61.5 (S) | 11.6 (S) | ✗ |
| | 464 | 47.35 Million (B) | 5166.1 (B) | 1843 (B) | ✗ |
| Fables | 152 | 39K | 7.1 | - | ✓ |

Table 2: Statistics of the datasets employed in this study. For the Fables dataset, alignments between sentence pairs are manually annotated.

it challenging to compare alignment results. To unify these metrics, we generate a distribution of similarity scores for a large set of unrelated text components. We then calculate the similarity score between two text components, $x$ and $y$ as follows:

$$S(x, y) = \sigma(Z(x, y)) - th_s) \times 2 - 1 \quad (3)$$

where $\sigma(.)$ is the logistic sigmoid function, $Z(x, y)$ is the z-score of the similarity between $x$ and $y$, and $th_s$ is a threshold for a positive score. This conversion standardizes the range of values for all scoring functions to between -1 and 1. The $th_s$ threshold ensures that pairs with a z-score less than $th_s$ receive a negative similarity score, indicating a possible lack of relatedness. This aligns with the requirements proposed by Dayhoff et al. (1978) for PAM matrices, one of the most widely used similarity metrics for protein sequence alignments which require the expected similarity score of two random unrelated components be negative so only related components get a positive similarity score.

The threshold value $th_s$ is crucial in distinguishing between similarity scores of semantically similar text pair from random text pair. We hypothesize similarity scores between unrelated text units follow a normal distribution. To determine $th_s$, we computed cosine similarities for 100M random paragraph pairs using SBERT embeddings. We use the fitted normal distribution ($\mu = 0.097, \sigma = 0.099$) presented in Figure 5 in the appendix to estimate the probability of chance similarity score $X$. We set a default minimum threshold of +3 z-score for SW alignment, resulting in a $\sim 0.0015\%$ probability of positive score for unrelated paragraphs following the three-sigma rule (Pukelsheim, 1994).

## 4 Statistical Significance of Alignments

Determining the significance of a semantic text alignment is a critical aspect that is dependent on both the domain and the user's definition of significance. For instance, two essays on the same topic are expected to have more semantically similar text

segments than two fictional books by different authors. We discuss how SW can produce weak noise alignments between unrelated texts in Appendix C. Significance testing can distinguish between real and noise alignments that occur by chance.

### 4.1 Computing Statistical Significance

We propose a sampling-based method for computing the statistical significance of alignment scores that can aid in computing thresholds for significance. Altschul and Gish (1996) hypothesizes that the gapped alignment scores of unrelated protein sequences follow an extreme value distribution of type I, specifically the Gumbel distribution. We argue that the same hypothesis holds for unrelated narrative texts sequences too and evaluate it here. Altschul and Gish (1996) defines the alignment score as the maximum value in the DP matrix computed by SW alignment. They estimate the probability of getting an alignment score $S \geq x$ as:

$$P(S \geq x) = 1 - \exp(-Kmne^{-\lambda x}) \quad (4)$$

Here $m$ and $n$ are lengths of the two random sequences, while $\lambda$ and $K$ are parameters that define the Gumbel distribution. The probability density function of the Gumbel distribution is:

$$f_{\text{gumbel}}(x) = \frac{1}{\beta} \times e^{-(z + e^{-z})} \quad (5)$$

where, $z = \frac{x-\mu}{\beta}$, $\mu = \log(K * m * n)/\lambda$ and $\beta = \frac{1}{\lambda}$. To estimate distribution parameters for the literary domain, we pair 1000 unrelated books by unique authors and align all possible pairs using SW with SBERT similarity at the paragraph level. After excluding the pairs having no alignment we get a distribution of size $\sim 2.5 \times 10^5$.

To simplify the estimation, we set $m$ and $n$ to be the mean length of books in the dataset and do maximum likelihood estimation to find the two unknown parameters $K$ and $\lambda$ for fitting the Gumbel distribution. We present the distribution of alignment scores and the associated Gumbel distribution

in Figure 2. We observe that the fitted Gumbel distribution closely follows the original distribution. We present p-values of some example alignments using the Gumbel distribution in Appendix B.

# 5 Experiments and Applications

## 5.1 Datasets

We created four different datasets for experiments on narrative alignment tasks over different scales and applications. The properties of these datasets, discussed below, are summarized in Table 2:

**RelBook Dataset:** Project Gutenberg (Gutenberg, n.d.)hosts tens of thousands of books that have been used in previous research in NLP (Rae et al., 2019). We manually selected a set of 36 non-English books that have multiple English translations in Project Gutenberg. We pair these translations to create a set of 36 pairs of related books.

**Classics Stories Dataset:** We extracted 14 story pairs from two books: *Tales From Dickens* by Hallie Erminie Rives, containing shorter adaptations of Dickens' classics for young readers, and *The World's Greatest Books — Volume 03*, containing selected abridged excerpts from Dickens' classics. Alignment of two abridged versions of *Oliver Twist* are shown in Table 1.

**SummaryBook Dataset:** We use summaries of over 2,000 classic novels from *Masterplots* (Magill, 1989). We intersected the *Masterplots* summaries with the set of books from Project Gutenberg, using titles, authors, and other metadata, to obtain a set of 464 pairs of books and summaries.

**Fables Dataset:** We curated seven different compilations of *Aesop's Fables* from Project Gutenberg, each containing independent 5-to-15 sentence versions of the classic fables. From this we created a set of 150 story pairs, each comprising of two different versions of the same fable. Two human annotators also manually aligned corresponding sentence pairs for each fable-pair, where many-to-many mapping is allowed. We found a $80.42\%$ agreement and a Cohen's Kappa score of $0.746$, indicating a substantial agreement (Landis and Koch, 1977) between the two annotators. Appendix D details the annotation process.

## 5.2 Comparison of Similarity Functions

In this experiment, we explore the effectiveness of different similarity metrics in distinguishing between related and unrelated book pairs. We hypothesize that related pairs (*RelBooks*) should yield

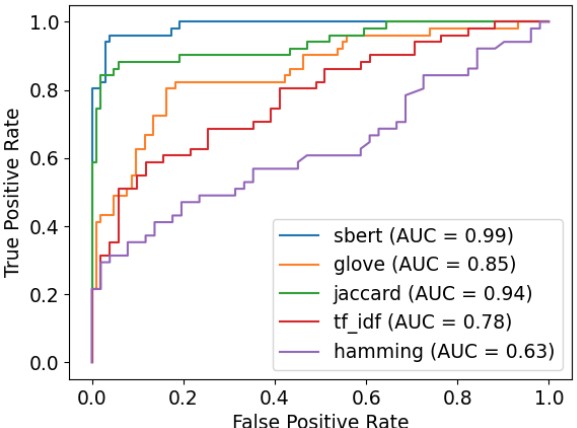

Figure 3: ROC curves distinguishing related and unrelated pairs of books from alignment order correlation, for different similarity metrics. SBERT is the best at identifying pairs of related books.

higher alignment scores than unrelated book pairs. We compute SW alignment for each related and unrelated book pair using the similarity metrics discussed in Section 3.2, and plot an ROC curve in Figure 3 to evaluate SW alignment performance for classifying relatedness.

We observe that the SBERT embeddings have the highest AUC score of $0.99$, with Jaccard coming close second with $0.94$. The weaker performance of widely-used metrics such as TF-IDF with $AUC = 0.78$ shows the non-triviality of the task as well as the strengths of the SBERT embeddings. We discuss how we create the unrelated book pairs and present an additional experiment using this data in Appendix C.

## 5.3 Plagiarism Detection for PAN13 Dataset

The PAN13 plagiarism detection dataset (Potthast et al., 2013) contains 10,000 pairs of documents from the ClueWeb 2009 corpus evenly split as training and test set. In each pair, text segments from the source document are inserted into the target document using automated obfuscation strategies, creating what are referred to as plagiarized cases. These cases can be identified through high-scoring local alignments of the two documents.

We use our alignment system at the sentence and paragraph level for this dataset and report the results in Table 3. We identify the optimal z-threshold $th_s$ for positive matches as discussed in the Section 3.3 using the training set and run the system against the test set. We do no optimization of our system, except for the z-threshold search. Our out-of-the-box system demonstrated competitive

| | No Obfuscation | | | Random Obfuscation | | | Translation Obfuscation | | | Summary Obfuscation | | | Entire Corpus | | |
|---|---|---|---|---|---|---|---|---|---|---|---|---|---|---|---|
| | precision | recall | F-1 | precision | recall | F-1 | precision | recall | F-1 | precision | recall | F-1 | precision | recall | F-1 |
| Torrejón and Ramos (2013) | .90 | .95 | **.92** | .91 | .63 | .74 | **.90** | .81 | .85 | .91 | .22 | .35 | **.89** | .76 | .82 |
| Leilei et al. (2013) | .76 | .91 | .83 | .86 | .79 | .82 | .86 | .85 | .85 | .96 | .30 | .46 | .83 | .81 | .82 |
| Suchomel et al. (2013) | .69 | **.99** | .81 | .83 | .69 | .75 | .68 | .67 | .67 | .67 | .56 | .61 | .73 | .77 | .75 |
| Sanchez-Perez et al. (2014) | .83 | .98 | .90 | .91 | .86 | **.88** | .88 | **.89** | **.88** | **.99** | .41 | .58 | .88 | **.88** | **.88** |
| Altheneyan and Menai (2020) | - | - | - | **.92** | .85 | **.88** | - | - | - | - | - | - | .89 | .85 | .87 |
| Jaccard+SW | .70 | .97 | .81 | .82 | .74 | .78 | .74 | .80 | .77 | .39 | .68 | .50 | .73 | .82 | .77 |
| SBERT+SW | **.91** | .79 | .84 | .89 | .43 | .58 | .86 | .66 | .75 | .93 | **.78** | **.85** | .84 | .67 | .75 |

Table 3: We report the performance of the top-3 teams in the plagiarism detection contest and recent state-of-the-art results, along with our text alignment tool on different subsets of the PAN-13 dataset (Potthast et al., 2013). The subsets are created based on how the plagiarism was inserted in the query documents. We present a subset of the results for Altheneyan and Menai (2020), as they did not provide results for all subsets of the dataset individually. SBERT+SW performs remarkably better than other methods on the Summary Obfuscation subset.

| Method | Unit Size | Percent Fidelity | Mean Rank | MRR | Worst Rank |
|---|---|---|---|---|---|
| Jaccard + SW | Sentence | 78.2 | 1.84 | 0.85 | 26 |
| | Paragraph | 77.7 | 2.36 | 0.84 | 41 |
| | Chunk | 21.4 | 6.36 | 0.35 | **18** |
| SBERT + SW | Sentence | 85.8 | 1.42 | 0.91 | 29 |
| | Paragraph | 88.4 | 1.36 | 0.93 | 36 |
| | Chunk | **90.6** | **1.31** | **0.94** | 34 |

Table 4: Performance of SBERT+SW and Jaccard+SW for different segmentation unit sizes for the BookSummary dataset. SBERT outperforms Jaccard for all unit sizes. SBERT shows better performance as summary sentences are aligned with larger units of the book. Percent fidelity denotes the fraction of cases where the related pair was ranked 1.

| Method | Precision | Recall | F1 |
|---|---|---|---|
| Random | 0.17 | 0.27 | 0.21 |
| Seq. Baseline | 0.31 | 0.54 | 0.40 |
| ChatGPT | 0.42 | 0.51 | 0.46 |
| SW+Jaccard | 0.62 | 0.69 | 0.65 |
| SW+SBERT | **0.63** | **0.71** | **0.67** |

Table 5: Performance of GNAT vs. ChatGPT for the Fables dataset. The test dataset consists of 2175 sentences for 120 fable pairs. Each sentence of the first fable of the pair is aligned with a sentence of the second fable or marked as unaligned.

performance against carefully-tuned approaches discussed by Potthast et al. (2013). Notably, our alignment method using SBERT embeddings outperformed other methods for detecting summary obfuscations, where the entire source document is summarized and placed at a random position in the target document. This superior performance can be attributed to SBERT capturing semantic similarities between text chunks of varying lengths.

## 5.4 Book/Summary Alignment

Here we evaluate the importance of segmentation size and the strength of different similarity metrics in aligning texts of different lengths. For each summary in the book-summary dataset, we pick 49 unrelated books of similar length creating 49 unrelated and 1 related pairs for each summary. We then compute the alignments, and rank the 50 pairs based on the maximum alignment score. We segment the summaries into sentences and repeat the experiment for different choices of segmentation sizes for the book: sentences, paragraphs, book chunks. For book chunks, we segment the book into $m$ equal chunks for a summary of length $m$. Table 4 reports the Mean Reciprocal Ranks (MRR)

and other results showing that SBERT is more effective in representing texts of different length for alignment than Jaccard.

## 5.5 Generative One-shot Learner Comparison

Generative models have shown promising results for many in-context learning tasks, where the model generates outputs based on one or more input-output examples. To evaluate performance of ChatGPT, based on GPT 3.5 architecture, (OpenAI, 2023) for text alignment, we provide the model with one example alignment prompt using the Fables dataset and then query it. The details of the prompts is discussed in Appendix Table 8. The limitation on the text size that can be passed as prompt to a generative model prevents longer texts from being used for this experiment.

We then compare the performance of ChatGPT and our tool in Table 5 against human annotation for the *Fables* dataset. Although ChatGPT outperformed our twin baselines of random and equally-spaced sequential alignment, our alignment method incorporating Jaccard and SBERT embeddings performed significantly better.

## 6 Conclusion

We have developed a tool for narrative text alignment using the Smith-Waterman algorithm with dif-

ferent similarity metrics, including a methodology to measure the statistical significance of aligned text segments. GNAT can be applied to the pairwise comparison of texts from any domain with varying unit sizes for text segmentation. While our evaluations were confined to English documents, the applicability of GNAT extends to languages supported by embedding models akin to SBERT. For languages with limited resources, embedding models can be trained following Reimers and Gurevych (2020), subsequently enabling the deployment of GNAT.

The next important direction here involves exploring how sequence database search tools like BLAST (Altschul et al., 1990) can be adapted for text search. One possible approach uses nearest neighbor search over embeddings in the database using parts of a query document to find seed positions for alignment, and then extends the alignments bidirectionally for faster heuristic searches. Another research direction is analyzing how the sampling based statistical significance testing method performs for different granularity levels, especially for alignment at the word or sentence level.

## Limitations

This paper documents the design and performance of a general narrative alignment tool, *GNAT*. Although we have presented experimental results to validate its utility in four different domains involving texts with varying absolute and relative lengths, every tool has practical performance limits.

As implemented, the Smith-Waterman algorithm uses running time and space (memory) which grows quadratically, as the product of the two text sizes. The recursive algorithmic approach of (Hirschberg, 1975) can reduce the memory requirements to linear in the lengths of the texts, at the cost of a constant-factor overhead in running time. Such methods would be preferred when aligning pairs of very long texts, although we have successfully aligned all attempted pairs of books over the course of this study.

The primary motivation of this project originates from a goal to create a general purpose alignment tool that anyone can use out-of-the-box with minimal effort. Almost all of our experiments showed SBERT embeddings to be superior to the other metrics. However creating SBERT embeddings is a computationally heavy task, especially when the text sequences are long and GPUs are unavailable. This restricts users with limited computational power from utilizing the full power of our tool. In such scenarios, using Jaccard as the choice of similarity metric will be ideal as Jaccard is the second strongest similarity metric supported by our tool and, it does not demand substantial computational resources. To address this limitation, we plan to include smaller versions of embedding models in the tool in the future that would require less resources.

## Acknowledgments

We would like to thank the anonymous reviewers for their valuable feedback, Arshit Jain (Stony Brook University) for development of the associated web tool, Dr. Naoya Inoue (Japan Advanced Institute of Science and Technology), Anshuman Funkwal, Pranave Sethuraj (Stony Brook University) for creating the Summary-Book dataset.

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

# Appendix

## A  Generating Multiple Local Alignments

The optimal Smith-Waterman alignment can be obtained by backtracing on the dynamic programming (DP) matrix, starting from the highest-scoring cell. To obtain the second-best local alignment, the initially aligned segments must be removed and the SW alignment DP matrix must be recomputed. But this is an inefficient method yielding a complexity of $O(q \times m \times n)$ to compute the $q$ best alignments.

Instead, we adopt a greedy approach to compute the alignments, where we backtrace from the current highest-scoring cell $(r_h, c_h)$ and terminate backtracing when we encounter a cell $(r_l, c_l)$ which either has a score of 0 or its row $r_l$ or column $c_l$ is already part of a previous alignment. Thus we get the alignment $a = (r_h, c_h, r_l + 1, c_l + 1)$. Every row $r$ such that $r_l < r \le r_h$ and every column $c$ such that $c_l < c \le c_h$ are marked as part of the current alignment. We then relaunch backtracing from the next highest-scoring cell. This method has a computational complexity of $O(mn \log(mn))$ as we sort all the cells once first. It yields the exact alignment for the best local alignment and approximations for the rest of the alignments.

## B  P-values of Chance Alignment

We investigate the probability of chance alignment by conducting an experiment involving various translations of *Captain Hatteras* by Jules Verne and *The Decameron* by Giovanni Boccaccio. We can reasonably expect the likelihood of obtaining alignments as strong as those observed by chance would exhibit an increasing trend across the following comparison scenarios in order:

- Translations of the same book by different translators.

- Translation of different parts of the same book by the same translator.

- Translation of different parts of the same book by different translators.

- Unrelated books.

We observe this pattern as anticipated in Table 6. To estimate these probabilities corresponding to the p-values, we utilize the cumulative distribution function (CDF) of the Gumbel distribution. We have fitted the parameters of the Gumbel distribution using alignment data of $2.5 \times 10^5$ unrelated pairs of books using the methodology discussed in Section 4.

## C  Local Alignments Can Capture Global Order

In this experiment we aim to investigate two phenomena of interest:

- How SW can pick up weak noise alignments even for unrelated pairs of texts. This strengthens the case for doing the statistical significance testing discussed in Section 4.

- The capability of SW to capture global order despite primarily operating at the local level, thereby showcasing its potential applicability in scenarios requiring global alignments.

SW produces local alignments which capture regions of strong similarity, but a collection of multiple local alignments are not necessarily sequential. It is quite possible that SW generates two alignments $a = \langle x_{ai}, x_{aj}, y_{ap}, y_{aq} \rangle$, and $b = \langle x_{bi}, x_{bj}, y_{bp}, y_{bq} \rangle$ where $(x_{ai} < x_{bi}) \wedge (y_{ap} > y_{bp})$. We hypothesize that local alignments of related pairs of books follow a similar sequence of events, and hence should be generally sequential. Conversely, alignments of unrelated book pairs should yield matches appearing in arbitrary order.

To test this hypothesis, we perform an experiment using 50 pairs of independently translated books from the RelBooks and the Classics Stories dataset (discussed in Section 5.1), plus 50 artificially constructed pairs where we pair one book from each pair in RelBooks and Classic Stories against a random book of similar length as the book they were originally paired with. As an example, the translations of the French novel *Around the World in Eighty Days* and the German novel *Siddhartha* form an unrelated pair.

We then align all 100 book pairs using SW, selecting the top 20 alignments with the highest scores for each of them, and then compute the correlation between $x_i$ and $y_p$ for these alignments. In Figure 4 we can observe that the unrelated pairs show weaker correlation than the related pairs. This clearly shows that SW can create some alignments for even unrelated text sequences, otherwise we would not have been able to extract the top 20 alignments for those pairs, underscoring the necessity of conducting statistical significance testing..

On the other hand, the high correlation values for majority of the related books demonstrate that SW can effectively capture the global sequence of events presented in the same order across both books.

## D Manual Annotation for *Fables* Dataset

The *Fables* dataset encompasses a collection of 150 fable pairs, comprising a total of 2175 sentences. Our annotation process involved the initial annotation of the entire dataset by a primary annotator. To enhance the reliability of the annotations, a second human annotator was engaged to annotate a randomly selected subset, accounting for 20% of the original dataset. The second set of annotation allowed us to gauge the subjectivity of the annotations and the level of difficulty of the task for both humans and automated systems. During experimentation, we only used the annotation of the primary annotator.

The annotators were presented with the pairs of fables, segmented into sentences, juxtaposed side-by-side. They were instructed to align each sentence of the first fable with a corresponding sentence from the second fable or mark it as unaligned. Notably, the first fable within each pair originated from the same book and typically featured a greater length compared to the fables from other books. The cumulative count of sentences in the first fable across all pairs amounted to 1369, while the second fable encompassed 806 sentences. The primary annotator aligned 769 sentences and marked the remaining 600 sentences as unaligned. For computing metrics presented in Table 5, we define aligned sentence pairs as positive examples and everything else as negative examples. Table 7 shows an example alignment created by the primary annotator for one pair of fables.

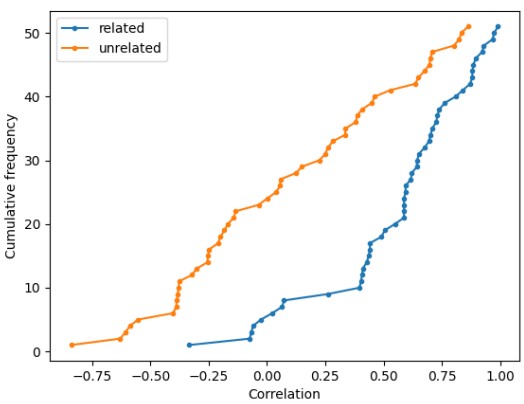

Figure 4: SW creates multiple local alignments instead of a single global alignment, but the events in two related books should typically maintain the sequential order. We measure sequential agreement as the correlation of paragraph number of *book1* and the number of the paragraph it aligns in *book2*. A K-S test reveals that the correlation value for alignments of related and unrelated pairs follow different distributions with statistically significant p-value = $2.08 \times 10^{-7}$. The effect size of the difference between the two distributions is calculated to be 1.19 using Cohen's d, indicating a large effect size.

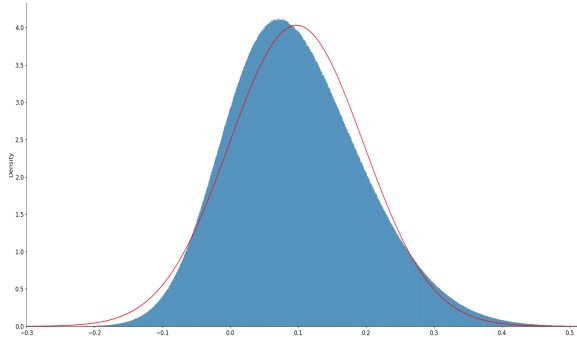

Figure 5: Distribution of SBERT cosine similarity scores between 100 million random pairs of paragraphs ($\mu = 0.097, \sigma = 0.099$), with the corresponding normal distribution (red curve). The CDF of the normal distribution is used as a proxy to estimate the probability of obtaining a similarity score $X$ by chance.

| Book1 | Book2 | Relation | P-value | Alignment Score |
|---|---|---|---|---|
| The Voyages and Adventures of Captain Hatteras (Translator: Osgood, J. R.) | The English at the North Pole Part I of the Adventures of Captain Hatteras (Translator: Anonymous, Gutenberg ID: 22759) | Same book, different translators | $1.85 \times 10^{-60}$ | 42.96 |
| The Voyages and Adventures of Captain Hatteras (Translator: Osgood, J. R.) | The Field of Ice Part II of the Adventures of Captain Hatteras (Publisher: Routledge, G) | Same book, different translators | $1.17 \times 10^{-49}$ | 35.42 |
| The Decameron Volume II (Translator: Rigg, J. M.) | The Decameron (Day 6 to Day 10) (Translator: Florio, J) | Same book, different translators | $6.02 \times 10^{-29}$ | 20.98 |
| The Decameron Volume I (Translator: Rigg, J. M.) | The Decameron (Day 1 to Day 5) (Translator: Florio, J) | Same book, different translators | $2.94 \times 10^{-23}$ | 17.01 |
| The Decameron Volume I (Translator: Rigg, J. M.) | The Decameron Volume II (Translator: Rigg, J. M.) | Different parts of same book, same translator | $2.19 \times 10^{-22}$ | 16.40 |
| The Decameron Volume I (Translator: Rigg, J. M.) | The Decameron (Day 6 to Day 10) (Translator: Florio, J) | Different parts of same book, different translator | $7.12 \times 10^{-18}$ | 13.25 |
| The English at the North Pole Part I of the Adventures of Captain Hatteras (Translator: Anonymous, Gutenberg ID: 22759) | The Field of Ice Part II of the Adventures of Captain Hatteras (Publisher: Routledge, G) | Different parts of same book, possibly different translator | $5.45 \times 10^{-3}$ | 2.87 |
| The Decameron Volume I (Translator: Rigg, J. M.) | The English at the North Pole Part I of the Adventures of Captain Hatteras (Translator: Anonymous, Gutenberg ID: 22759) | Unrelated books | $1.18 \times 10^{-1}$ | 1.92 |
| The Decameron Volume II (Translator: Rigg, J. M.) | The Field of Ice Part II of the Adventures of Captain Hatteras (Publisher: Routledge, G) | Unrelated books | $2.99 \times 10^{-1}$ | 1.60 |

Table 6: The p-value for finding alignment by chance goes up as the strength of relatedness between books decrease. The probabilities are calculated using the Gumbel distribution fitted using methodology discussed in Section 4. The null hypothesis in this context is that the alignments come from unrelated text sequences.

| Sentence # | Fable 1 | Alignment Annotation | Fable 2 |
|---|---|---|---|
| 0 | A Gnat flew over the meadow with much buzzing for so small a creature and settled on the tip of one of the horns of a Bull. | (0, 0) | A Gnat alighted on one of the horns of a Bull, and remained sitting there for a considerable time. |
| 1 | After he had rested a short time, he made ready to fly away. | (1, 1) | When it had rested sufficiently and was about to fly away, it said to the Bull, "Do you mind if I go now?" |
| 2 | But before he left he begged the Bull's pardon for having used his horn for a resting place. | (2, -1) | The Bull merely raised his eyes and remarked, without interest, "It's all one to me; I didn't notice when you came, and I shan't know when you go away. |
| 3 | "You must be very glad to have me go now," he said. | (3, 1) | "We may often be of more consequence in our own eyes than in the eyes of our neighbours." |
| 4 | "It's all the same to me," replied the Bull. | (4, 2) | |
| 5 | "I did not even know you were there." | (5, 2) | |
| 6 | _We are often of greater importance in our own eyes than in the eyes of our neighbor.__The smaller the mind the greater the conceit._ | (6, 3) | |

Table 7: Example alignment created by human annotator for the Aesop's fable *"The Gnat and the Bull"*. The pair $(2, -1)$ in the third column denotes that the annotator decided to let sentence 2 of Fable 1 remain unaligned.

| Prompt |
|---|
| Given two texts t1 and t2 in the form of numbered list of sentences, list the semantically similar sentence pairs using their numbers in the form (i,j) where 0 <= i < 7 and 0 <= j < 5 where the first sentence of the pair comes from t1 and the second sentence of the pair comes from t2. t1 has 7 sentences and t2 has 5 sentences. Do not include anything other than the list of pairs in your response.

t1:
0. A Boy was given permission to put his hand into a pitcher to get some filberts.
1. But he took such a great fistful that he could not draw his hand out again.
2. There he stood, unwilling to give up a single filbert and yet unable to get them all out at once.
3. Vexed and disappointed he began to cry.
4. "My boy," said his mother, "be satisfied with half the nuts you have taken and you will easily get your hand out.
5. Then perhaps you may have some more filberts some other time."
6. _Do not attempt too much at once._
t2:
0. A Boy put his hand into a jar of Filberts, and grasped as many as his fist could possibly hold.
1. But when he tried to pull it out again, he found he couldn't do so, for the neck of the jar was too small to allow of the passage of so large a handful.
2. Unwilling to lose his nuts but unable to withdraw his hand, he burst into tears.
3. A bystander, who saw where the trouble lay, said to him, "Come, my boy, don't be so greedy: be content with half the amount, and you'll be able to get your hand out without difficulty."
4. Do not attempt too much at once.

Output:
(0, 0)
(1, 1)
(2, 1)
(3, 2)
(4, 3)
(6, 4)

Given two texts t1 and t2 in the form of numbered list of sentences, list the semantically similar sentence pairs using their numbers in the form (i,j) where 0 <= i < 5 and 0 <= j < 3 where the first sentence of the pair comes from t1 and the second sentence of the pair comes from t2. t1 has 5 sentences and t2 has 3 sentences. Do not include anything other than the list of pairs in your response.

t1:
0. A Cock was busily scratching and scraping about to find something to eat for himself and his family, when he happened to turn up a precious jewel that had been lost by its owner.
1. "Aha!" said the Cock.
2. "No doubt you are very costly and he who lost you would give a great deal to find you.
3. But as for me, I would choose a single grain of barley corn before all the jewels in the world."
4. _Precious things are without value to those who cannot prize them._
t2:
0. A Cock, scratching the ground for something to eat, turned up a Jewel that had by chance been dropped there.
1. "Ho!" said he, "a fine thing you are, no doubt, and, had your owner found you, great would his joy have been.
2. But for me! give me a single grain of corn before all the jewels in the world."

Output: |

| ChatGPT Response | GNAT | Human Annotation |
|---|---|---|
| (0, 0) | (0, 0) | (0, 0) |
| (1, 1) | (1, 0) | (1, 1) |
| (2, 2) | (2, 1) | (2, 1) |
| (3, 2) | (3, 2) | (3, 2) |
| (4, 1) | | |

Table 8: In the prompt, we provide an example input pair of fables and their corresponding alignment following which we instruct ChatGPT to create alignment for the next pair of fables. The human annotation keeps sentence 4 of fable 1 unaligned but ChatGPT incorrectly aligns it with sentence 1 of fable 2. In the prompt, the example alignment also has an instance where the gold alignment keeps a sentence unaligned (sentence # 5). We allow GNAT to do many-to-many matching here by slightly modifying the recurrence in Equation 1.