# OpenReview forum: "GNAT: A General Narrative Alignment Tool"
_EMNLP/2023/Conference — EMNLP 2023 Main_

### Official Review · Reviewer_ynEf · 2023-08-03

**Soundness:** 4

**Excitement:**

4: Strong: This paper deepens the understanding of some phenomenon or lowers the barriers to an existing research direction.

**Missing References:**

No missing references. In fact, the number of references is quite large, maybe a little too large.


**Paper Topic And Main Contributions:**

The paper presents GNAT (for General Narrative Alignment Tool), a new algorithmic sequence alignment proposal for narrative alignment. The authors try to extend both the techniques (e.g. Smith-Waterman local alignment algorithm) and the rigour/formality of the sequence analysis techniques used in Bioinformatics (e.g. for gene-to-gene comparisons) to the case of general narrative texts. They have paid attention to the case of distant but semantically similar texts (e.g. translations), applicability when dealing not only with similar but dissimilar granularities (e.g. complete texts vs. summaries), also designing a methodology for computing the statistical significance of the text alignments obtained.


Their approach is empirically tested with 5 distinct distance metrics (cosine similarity on SBERT embeddings, Jaccard coefficient, TF-IDF, cosine similarity on GloVe Mean embeddings and Hamming Distance) on 4 distinct application domains (summary-to-book alignment, translated book alignment, plagiarism detection and short story alignment) and 4 datasets of different nature. In order to allow proper comparation between the results obtained with different scoring functions, a unification function is proposed.


The authors make a good and illustrative introduction to the task and its associated problems and defficiencies.

The Section "Related Work" is ok, since it covers the different aspects of this work as best as they can, taking into account the space limits they have to deal with.

Their major contributions are clearly stated, although their "primary" motivation, as stated in the Sect. "Limitations" is to create a general purpose alignment tool that anyone can use out-of-the-box with minimal effort.

The possibility of computing statistical significant scores on the resulting alignments is specially interesting. They propose a sampling-based method based on a Gumble distribution whose parameters were estimated empirically.

The results obtained are encouraging and seem to support the validity of their approach.

The tool and the evaluation datasets they have used will be publicly released.

**Questions For The Authors:**

Sect. 4.1 describes their sampling-based method for computing the statistical significance of algnments. To do that, they use a Gumbel distribution whose parameters were estimated empirically by pairing 1K unrelated books at the paragraph level. My question is: at what level this selection of granularities (books aligned a paragraph level) may influence the figures obtained if we are calculating the statistical significance of algning, for example, short texts at word or phrase level (i.e. completely different granularities).

**Reasons To Accept:**

Quite well-structured.
Well-written and easy to read (although a little more space would be of great help to concatenate the text properly).
Not novel by itself, but the authors apply existing concepts and techniques to a different context in a novel way.
The possibility of computing statistical significant scores for narrative alignment sounds great.
Wide evaluation experiments.

**Reasons To Reject:**

I see no special reasons for it.


**Reproducibility:**

3: Could reproduce the results with some difficulty. The settings of parameters are underspecified or subjectively determined; the training/evaluation data are not widely available.

**Reviewer Confidence:**

3: Pretty sure, but there's a chance I missed something. Although I have a good feel for this area in general, I did not carefully check the paper's details, e.g., the math, experimental design, or novelty.

**Typos Grammar Style And Presentation Improvements:**

* Sect. 1 "Introduction" is too long and goes beyond a mere introduction to the paper. The autors should re-structure and split it (for instance, from l. 113 on)
* Figure 1 is not referenced in the text.
* The same for Table 1.
* Sect. 3.1, l.310: you have not previously explained in the textwhat function H is.
* Sect. 3.1, l.313: missing point in "gap penalty However"
* Sect. 4.1, l.449: "gumbel" --> "Gumbel"
* Sect. 5.1, l.462-463: when introducing Project gutemberg, a footnote with its URL and last access date is enough, if not better. Don't add a bib. reference for it.
* caption of Figure 3: "corrleation" -> "correlation"

* Sect. "References":
     - some missed uppercases here and there
     - when using name initials (e.g. l.632, 683), you must homogeinize the use (or not) of .
     - "remains" of old copy-pastes in l.642, 857

---

> ### Author Rebuttal · Authors · 2023-08-28
>
> Thank you for your comprehensive review.
>
> We have not yet done experiments with smaller granularity levels for the statistical significance testing, such as measuring statistical significance of alignment at the word level between short texts. In theory, our sampling based method should still work in this case, but we estimate that shorter texts probably will yield less statistical power to correctly recognize distant alignments. If the paper is accepted, we will add some experiments/discussion in the camera-ready version on how different granularity levels affect the sampling based method's applicability. We thank you for bringing up the point of granularity level for statistical significance testing as we think it is an interesting question to explore.
>
> We agree with the presentation improvement suggestions provided by the reviewer. If the paper gets accepted, we will incorporate all the suggestions in the camera-ready version—and look forward to an additional page to do with!

---

### Official Review · Reviewer_jEsz · 2023-08-04

**Soundness:** 4

**Excitement:**

4: Strong: This paper deepens the understanding of some phenomenon or lowers the barriers to an existing research direction.

**Paper Topic And Main Contributions:**

There are exists different methods to identify similar segments between original documents, but it is difficult to find similarities between distant versions of narratives. The authors propose a tool for narrative text alignment based on the Smith-Waterman algorithm from bioinformatics with text similarity metrics called GNAT.
GNAT is a general narrative alignment tool can be applied to compare texts to find similar segments for any domain. The authors do experiments on four problem domains such as summary-to-book alignment, translated book alignment, short story alignment, and plagiarism detection.

**Reasons To Accept:**

The paper can be accepted for the following reasons: the authors offer a tool that anyone can use without effort and evaluate it on the four different domains. The research lies on the intersection of bioinformatics and text similarity metrics from NLP. The title and abstract are correctly selected and written. The paper is correctly structured, easy to follow. Pictures and tables are relevant and clearly explained.

**Reasons To Reject:**

GNAT requires a computational power; therefore, it may restrict users (especially when GPU unavailable) to use this tool.

**Reproducibility:**

5: Could easily reproduce the results.

**Reviewer Confidence:**

3: Pretty sure, but there's a chance I missed something. Although I have a good feel for this area in general, I did not carefully check the paper's details, e.g., the math, experimental design, or novelty.

---

> ### Author Rebuttal · Authors · 2023-08-28
>
> Thank you for the favorable review. Even though no direct questions were given, we briefly address the point of GNAT requiring computational power.
>
> For small texts, SBERT will be slightly slower on CPU but still can be worked with. For longer texts like classic books, we plan to keep embeddings stored on our side for the web tool lowering the burden on the user side. In any other case, Jaccard similarity always remains a strong candidate in low resource scenarios.   Indeed, we intend to provide for the costs of maintaining an online web server to support interested users experimenting with such computations using CPU, in addition to making the code available so the user can run on their end using GPU as needed.

---

### Official Review · Reviewer_Abc9 · 2023-08-04

**Soundness:** 4

**Excitement:**

4: Strong: This paper deepens the understanding of some phenomenon or lowers the barriers to an existing research direction.

**Missing References:**

N/A

**Paper Topic And Main Contributions:**

The paper presents an extensive collection of experiments on the
question of automatic text alignments for text pairs under different
task contexts.  The authors created annotated data for the different
tasks an d compared a number of alignment techniques on them.
The tools and data will be public.

**Questions For The Authors:**

* how easy is it to extend this work to other languages, particularly languages with more complex
and rich morphology? can you elaborate on generalizability to other languages?


**Reasons To Accept:**

* well written paper.
* interesting problem applied to a number of domains/tasks
* tools will be public
* data will be public


**Reasons To Reject:**

Hard to find a reason. Perhaps, it is unclear how well this work generalizes beyond English?

**Reproducibility:**

3: Could reproduce the results with some difficulty. The settings of parameters are underspecified or subjectively determined; the training/evaluation data are not widely available.

**Reviewer Confidence:**

3: Pretty sure, but there's a chance I missed something. Although I have a good feel for this area in general, I did not carefully check the paper's details, e.g., the math, experimental design, or novelty.

**Typos Grammar Style And Presentation Improvements:**

Missing period:
313  gap penalty However,

---

> ### Author Rebuttal · Authors · 2023-08-28
>
> Thank you for your detailed and favorable review.
>
> The generalizability to other languages is indeed an interesting research direction we intend to explore in future. The two main components are the core alignment algorithm, Smith-Waterman (SW) in our case, and the similarity metric employed by SW. The core alignment algorithm can be applied to any language without any modification as we show how SW can work for both the bioinformatics and the NLP domain.
>
> Thus the similarity metrics are the challenging part for alignment in other languages. The strongest metric we explored, SBERT embeddings, require a similar model for the specific language we want to align text from.  SBERT models for popular languages other than English are available in Huggingface (e.g., KR-SBERT-V40K-klueNLI-augSTS for Korean, sbert-cased-finnish-paraphrase for Finnish, etc). For languages with limited resources and no pre-trained models, "Making Monolingual Sentence Embeddings Multilingual using Knowledge Distillation" by Reimer and Gurevych can be an excellent starting point. This rationale also holds true for the GloVe Mean Embedding approach we employed.
>
> For the other three metrics, a simple way can be to extract the root/stem of each individual word and compare them. We believe these should suffice for simple alignments.  We will try to add some small/qualitative experiments on this issue for the camera-ready version of the paper upon acceptance.
>
> To summarize, applying GNAT for the most popular languages should be straight-forward when SBERT or similar embedding models are available. There are fairly simple ways for modifying the algorithm for applying it to morphologically rich languages with limited resources. but more experiments need to be performed to ascertain the degree of generalizability of GNAT as a tool for low-resource languages.

---

### Meta-Review · Area_Chair_CNWw · 2023-09-18

**Recommendation:** 5

**Metareview:**

The reviewers agree that the paper is sound and exciting. Several reviewers mention the paper's writing is easy to understand and the paper is generally well-written. The paper promises to release its tools and data, which will help other researchers reproduce and build upon this work. Reviewers generally do not find issues with the paper. There was one concern about computational requirements, but the authors addressed the concerns during the author response period.

---

### Decision · Program_Chairs · 2023-10-07

**Decision:**

Accept-Main

**Comment:**

The reviewers agree that the paper is sound and exciting. Several reviewers mention the paper's writing is easy to understand and the paper is generally well-written. The paper promises to release its tools and data, which will help other researchers reproduce and build upon this work. Reviewers generally do not find issues with the paper. There was one concern about computational requirements, but the authors addressed the concerns during the author response period.